# Generation of vascularized brain organoids to study neurovascular interactions

Xin-Yao Sun[1,2,3†], Xiang-Chun Ju[2*†‡], Yang Li[1], Peng-Ming Zeng[1], Jian Wu[1], Ying-Ying Zhou[2,3], Li-Bing Shen[2], Jian Dong[1], Yue-Jun Chen[2,3], Zhen-Ge Luo[1*]

[1]School of Life Science and Technology, ShanghaiTech University, Shanghai, China; [2]Institute of Neuroscience, Center for Excellence in Brain Science and Intelligence Technology, Chinese Academy of Sciences, Shanghai, China; [3]University of Chinese Academy of Sciences, Beijing, China

**Abstract** Brain organoids have been used to recapitulate the processes of brain development and related diseases. However, the lack of vasculatures, which regulate neurogenesis and brain disorders, limits the utility of brain organoids. In this study, we induced vessel and brain organoids, respectively, and then fused two types of organoids together to obtain vascularized brain organoids. The fused brain organoids were engrafted with robust vascular network-like structures and exhibited increased number of neural progenitors, in line with the possibility that vessels regulate neural development. Fusion organoids also contained functional blood–brain barrier-like structures, as well as microglial cells, a specific population of immune cells in the brain. The incorporated microglia responded actively to immune stimuli to the fused brain organoids and showed ability of engulfing synapses. Thus, the fusion organoids established in this study allow modeling interactions between the neuronal and non-neuronal components in vitro, particularly the vasculature and microglia niche.

**\*For correspondence:**
xiangchun.ju@oist.jp (X-CJ);
luozhg@shanghaitech.edu.cn
(Z-GL)

†These authors contributed equally to this work

**Present address:** ‡Okinawa Institute of Science and Technology, Onna-son, Japan

**Competing interest:** The authors declare that no competing interests exist.

## Editor's evaluation

This article puts forward a new approach to generate vascularized brain organoids. The novelty of their approach lies in the simultaneous production of vessel-like networks and brain-resident microglia immune cells in a single organoid, and data demonstrating that the vessels are patent to allow fluid flow when pressurized fluid is delivered to the vascular tube. The fusion of brain and vessel organoids resulted in robust engraftment of vessel-like structures and microglia around ventricular zone-like structures, correlating with increased neuronal progenitors.

## Introduction

Recently, brain organoids (BOrs) derived from human pluripotent stem cells (PSCs), including induced PSCs (iPSCs) and embryonic stem cells (ESCs), have been developed to model developmental programs of human fetal brain, recapitulate developmental, psychiatric, and degenerative brain diseases (*Amin and Paşca, 2018*; *Di Lullo and Kriegstein, 2017*; *Kelava and Lancaster, 2016*; *Lancaster and Knoblich, 2014*). However, the lack of neurovascular system, which is not only required for oxygen and nutrient supply, but also regulates neurogenesis and brain functions (*Delgado et al., 2014*; *Tata et al., 2016*; *Zhao et al., 2015*; *Zlokovic, 2011*), limits the applications of brain organoids. Thus, vascularization of brain organoids represents one of the most demanded improvements in the field (*Di Lullo and Kriegstein, 2017*; *Giandomenico and Lancaster, 2017*; *Kelava and Lancaster, 2016*; *Mansour et al., 2021*).

**eLife digest** Understanding how the organs form and how their cells behave is essential to finding the causes and treatment for developmental disorders, as well as understanding certain diseases. However, studying most organs in live animals or humans is technically difficult, expensive and invasive. To address this issue, scientists have developed models called 'organoids' that recapitulate the development of organs using stem cells in the lab. These models are easier to study and manipulate than the live organs.

Brain organoids have been used to recapitulate brain formation as well as developmental, degenerative and psychiatric brain conditions such as microcephaly, autism and Alzheimer's disease. However, these brain organoids lack the vasculature (the network of blood vessels) that supplies a live brain with nutrients and regulates its development, and which has important roles in brain disorders. Partly due to this lack of blood vessels, brain organoids also do not develop a blood brain barrier, the structure that prevents certain contents of the blood, including pathogens, toxins and even certain drugs from entering the brain. These characteristics limit the utility of existing brain organoids.

To overcome these limitations, Sun, Ju et al. developed brain organoids and blood vessel organoids independently, and then fused them together to obtain vascularized brain organoids. These fusion organoids developed a robust network of blood vessels that was well integrated with the brain cells, and produced more neural cell precursors than brain organoids that had not been fused. This result is consistent with the idea that blood vessels can regulate brain development.

Analyzing the fusion organoids revealed that they contain structures similar to the blood-brain barrier, as well as microglial cells (immune cells specific to the brain). When exposed to lipopolysaccharide – a component of the cell wall of certain bacteria – these cells responded by initiating an immune response in the fusion organoids. Notably, the microglial cells were also able to engulf connections between brain cells, a process necessary for the brain to develop the correct structures and work normally.

Sun, Ju et al. have developed a new organoid system that will be of broad interest to researchers studying interactions between the brain and the circulatory system. The development of brain-blood-barrier-like structures in the fusion organoids could also facilitate the development of drugs that can cross this barrier, making it easier to treat certain conditions that affect the brain. Refining this model to allow the fusion organoids to grow for longer times in the lab, and adding blood flow to the system will be the next steps to establish this system.

Blood vessels of the vertebrate brain are formed via sequential vasculogenesis and angiogenesis processes, which involve the initial invasion of endothelial cells (ECs) into the neuroepithelium regions via the perivascular plexus, their subsequent coalescence into primitive blood vessels, and growth and remodeling to form a mature vascular network (*Lee et al., 2009*). ECs are derived from the mesoderm-derived angioblasts (*Zadeh and Guha, 2003*). It has been shown that ECs can be derived in vitro from human PSCs, and these ECs can be potentially useful in engineering artificial functional blood vessels (*Harding et al., 2017*). However, the generation of complex vascularized organs from PSCs is still challenging because it depends on the exquisite orchestration of cues from multiple germ layers and the gene expression profiles of ECs are controlled by finely patterned microenvironmental cues during organogenesis (*Cleaver and Melton, 2003*). Recently, Wimmer et al. reported the generation of self-organizing human blood vessel organoids (VOrs) induced from PSCs and the application in the study of diabetic vasculopathy (*Wimmer et al., 2019*). Given the mesodermal origin of ECs and ectodermic origin of neural fates (*Nostro et al., 2008*; *Stern, 2005*), one barrier for the generation of vascularized BOrs is the difficulty in simultaneous application of induction factors for distinct germ layers and cell fates due to mutual repression.

Since the early attempt to vascularize the BOrs by embedding with iPSCs induced ECs (*Pham et al., 2018*), several additional strategies have been developed. One strategy took advantage of natural angiogenesis of host blood vessels that sprout and grow into the grafted cerebral organoids, which later exhibited lower cell death rate and enhanced maturation (*Mansour et al., 2018*). Another approach utilized transcription factor-mediated differentiation of a subset of PSCs into EC-like cells during cerebral organoid induction, whose maturation process was also enhanced (*Cakir et al., 2019*).

Another study has tried co-culture with ECs or their progenitors during cerebral organoid formation (**Shi et al., 2020**). Although grafted BOrs appeared to have established functional blood vessels (**Mansour et al., 2018**), none of these methods can form an entirely and integrated vascular network in the cerebral organoids in vitro. They all lacked the functional microglial cells, the only lifelong resident immune cells, which are derived from mesodermal origin (**Muffat et al., 2016**). In addition, the blood–brain barrier (BBB), the structure mainly composed of ECs, astrocytes, and pericytes, which protects the brain from circulation, is also lacking in the current ectodermal BOr models. By co-culture of primary ECs, pericytes, and astrocytes, the BBB spheroids were created as an in vitro screening platform for brain-penetrating agents (**Bergmann et al., 2018**; **Cho et al., 2017**).

Here, we develop an induction approach for brain-specific vascular organoids, which were cultured in medium containing neurotrophic factors at the maturation stage, to obtain cerebrovascular characteristics of the VOrs. Interestingly, a large number of microglial cells were induced by this approach along with other types of vascular cells. The VOrs were then fused with the cerebral organoids in the Matrigel, leading to the formation of vascularized BOrs with invasion of microglia, which could be activated upon immune stimuli. Thus, this study invents an advanced strategy that incorporates vascular and microglia into BOrs, providing a platform for the study of interactions between neuronal and non-neuronal components during brain development and functioning.

## Results

### Generation of the VOrs

It has been shown that the canonical Wnt signaling is required for the development of ESC-derived mesoderm (**Lindsley et al., 2006**) and the activation of Wnt signaling induces the mesoderm differentiation from human PSCs (**Nostro et al., 2008**). Considering that EC-generating vascular progenitors (VPs) are derived from mesoderm during embryogenesis (**Gupta et al., 2006**), we performed guided mesodermal induction of H9 human embryonic stem cells (hESCs), followed by endothelial differentiation. First, we treated 2-day-old (D2) embryonic bodies (EBs) from hESCs, which stably expressed GFP, with GSK3 inhibitor CHIR99021 to activate the canonical Wnt signaling for mesoderm induction (**Figure 1A**). After 2 days, the EBs were treated with basic fibroblast growth factor (bFGF), vascular endothelial growth factor (VEGF), and bone morphogenetic protein 4 (BMP4), all of which have been shown to be able to promote VP differentiation into ECs (**Cai et al., 2012**; **Jih et al., 2001**). After 3 days, the differentiated ECs were incubated with endothelial medium ECGM-MV2 (MV2 hereafter) containing VEGF for 5 days for further maturation, and then embedded in Matrigel droplets. At late maturation stages, neurotropic reagents N2 and B27 were added into the maturation medium, which presumably might be able to induce some cell types with specific brain vessel features (**Figure 1A**). Notably, vessel-like structures sprouted out from the spheroids at day 16 (D16) (**Figure 1B**), reminiscent of initial vasculogenesis and angiogenesis. Remarkably, the VOrs showed gradual increase in the size during the maturation stage after day 16 (D16), with apparent tubular network characteristics at D20 (**Figure 1B**, **Figure 1—figure supplement 1A**). At D40, the VOr gradually formed into a sphere with smooth edges due to Matrigel's wrapping, and its internal blood vessel structures developed more dense and integrated (**Figure 1—figure supplement 1B**).

To verify the cell fates in developing VOrs, we performed quantitative PCR to determine the expression of stemness or vascular-specific genes at different time upon organoid differentiation. As shown in **Figure 1C**, the stemness markers (*NANOG, OCT4*) showed marked decrease 2 days upon mesoderm induction (D4), whereas the vessel markers (*PECAM1, VE-cadherin, VWF, VEGFR1, VEGFR2,* and *PDGFRβ*) markedly increased after VP differentiation (D7 and thereafter). In line with this, flow cytometry results revealed the appearance of GFP⁺CD31⁺ ECs after D7, indicating the induction of the ECs (**Figure 1—figure supplement 1C**). The relative reduction in the proportion of ECs in later stages might be due to the appearance of other cell types, such as fibroblasts, pericytes, and smooth muscle cells.

Morphologically, CD31⁺ ECs in VOrs at D40 showed integrated and complex structures (**Figure 1D**) and exhibited remarkable vascular branches and tips undergoing angiogenesis-like processes (**Figure 1E**, **Figure 1—figure supplement 1D**). The 3D-reconstructed cross-section revealed tubular structures in VOrs, reminiscent of vessel lumen (**Figure 1E**, see top right e1). To determine the connectivity and integrity of vessel-like structures, we injected fluid into the lumens and observed that the

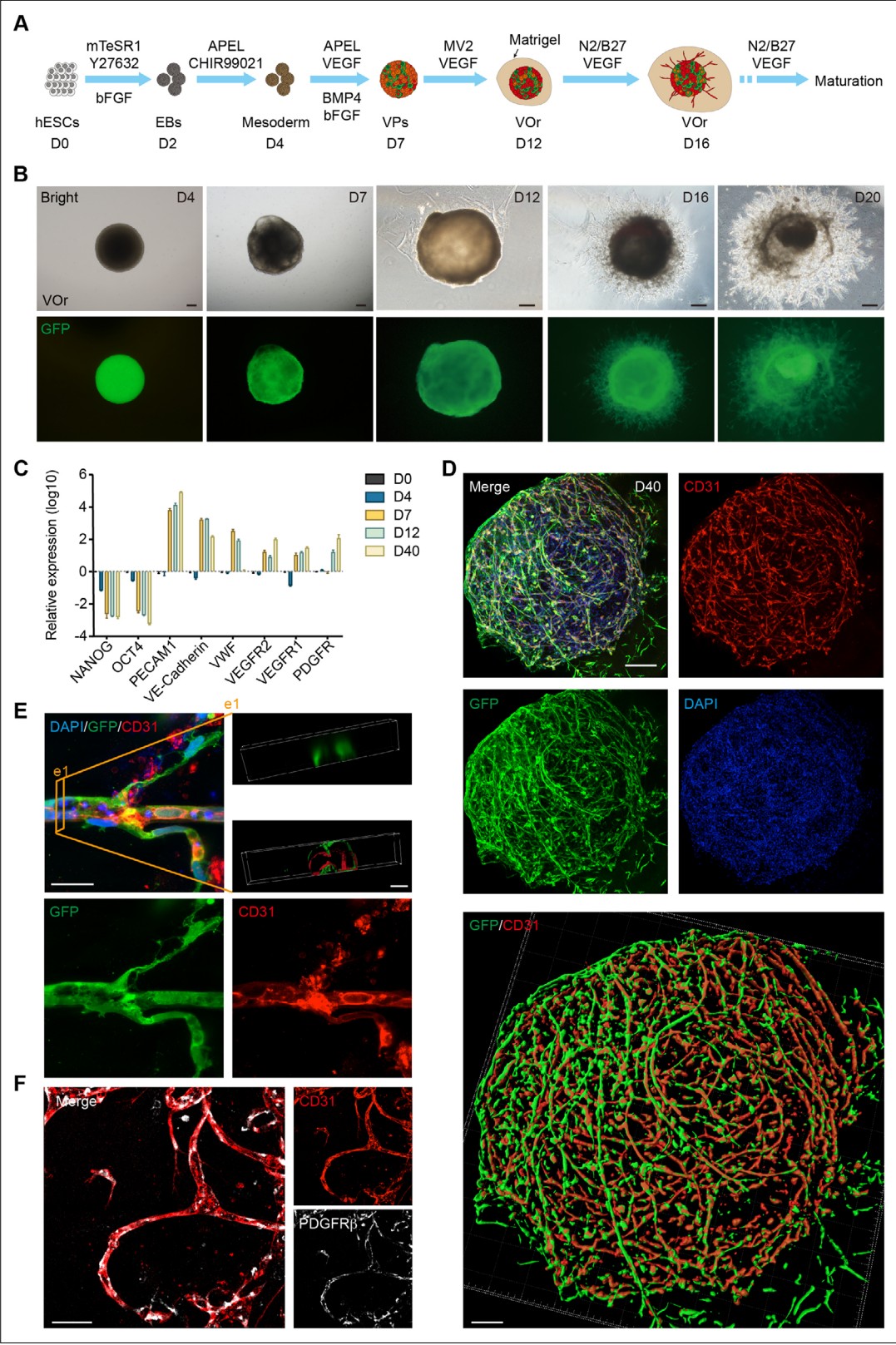

**Figure 1.** Generation of an in vitro model of vessel organoids (VOr). (**A**) Schematic view of the methods for generating VOrs from GFP-hESC. EBs: embryonic bodies; VPs: vascular progenitors; VOr: vessel organoid; hESC: human embryonic stem cell. (**B**) Different developmental stages of VOrs from day (D) 4 to D20. Top, right field; bottom, GFP. Scale bar, 200 μm. (**C**) qPCR analysis for expression of stem markers (*NANOG, OCT4*) and

*Figure 1 continued on next page*

*Figure 1 continued*

vessel markers (*PECAM1, VE-Cadherin, VWF, VEGFR1, VEGFR2, PDGFR*) in developing VORs, using GAPDH as internal control. Data are presented as mean ± SEM (n = 3 independent experiments), error bars indicate SEM. (**D**) Immunostaining of GFP and CD31 in D40 VORs. Scale bar, 200 μm. Bottom: Imaris reconstruction of VORs showing integrated vasculature structures. (**E**) Immunostaining of GFP and CD31 for the vascular structures in VORs. Scale bar, 20 μm. Top right: section view in VOr showing the lumen structure. (**F**) Immunostaining of CD31 and PDGFRβ for endothelial cells and pericytes, respectively. Scale bar, 50 μm.

The online version of this article includes the following video and figure supplement(s) for figure 1:

**Figure supplement 1.** Vessel organoids (VORs) recapitulate human vessel development.

**Figure 1—video 1.** Phosphate-buffered saline (PBS) fluid was microinjected into the vessel-like lumen in day (D) 40 vessel organoid (VOr) with continuous pressure, showing liquid flow and vessel wall expansion without leakage.

https://elifesciences.org/articles/76707/figures#fig1video1

hydraulic pressure caused liquid flow and vessel wall expansion without leakage (see *Figure 1—video 1*). Notably, PDGFRβ-labeled pericytes that are believed to regulate EC maturation, stabilize vessel wall, and control angiogenesis were also observed in close contact with ECs undergoing vessel differentiation (*Figure 1F*). For more details about the vasculature morphology, we used the Angiotool software that had been used as a tool for quantitative vessel analysis (*Zudaire et al., 2011*). The average vessel length was around 400 mm, the vessel lacunarity was 0.15, and the total number of junctions was about 700–800 per VOr (*Figure 1—figure supplement 1E*).

To assess the function of ECs in VORs, we determined the ability to incorporate DiI-acetylated low-density lipoprotein (DiI-Ac-LDL), as shown in a previous study (*Lehle et al., 2016*). We found that VORs after D14 already had the ability of uptaking DiI-Ac-LDL, whereas ESCs could not (*Figure 1—figure supplement 1F*). Thus, we have successfully established a fully structured and functional vessel organoid model.

## Cell composition of brain-specific VORs resembles brain vessels in vivo

The vessel system of the brain contains a variety of vascular cell types (*Vanlandewijck et al., 2018*). To investigate the fidelity of VORs in recapitulating the cerebrovascular cell types, we performed single-cell RNA sequencing (scRNA-seq) of VORs at D40 using 10x Genomic chromium system (*Macosko et al., 2015*; *Zilionis et al., 2017*). After the quality control data filtering, we analyzed transcriptome of about 7000 single cells, with 200–7000 genes detected per cell and the mitochondrial gene ratio under 5%. The mean reads per cell of two batches of independent samples were highly correlated (*Figure 2—figure supplement 1A*), indicating negligible batch variance. According to cell-type markers of the mice brain vessels identified by single-cell sequencing (*He et al., 2018*; *Vanlandewijck et al., 2018*), the cells in VORs were clustered into nine main cell types (*Figure 2A*), including fibroblast (FB), pericyte (PC), proliferative vascular progenitor (MKI67$^+$ VP), EC, smooth muscle cell (SMC), microglia (MG), immune cell (IM), and unknown cluster (*Figure 2B*). FB accounted for the highest proportion of total cells and MG the lowest (*Figure 2—figure supplement 1B*). The proportion of ECs was in line with the flow cytometry results at D40 (*Figure 1—figure supplement 1C*). We chose the top five highly expressed marker genes of each cluster (*Figure 2—figure supplement 1C*) and analyzed their expression patterns in each cell type (*Figure 2C*). Immunostaining showed that the vasculatures in VORs exhibited positive signals of the SMC marker αSMA, the pericyte marker PDGFRβ, and the EC marker CD31 (*Figure 2D and E*), confirming the results obtained using scRNA-seq. Meanwhile, the presence of MG-like cells was verified by the staining with specific markers IBA1, TREM2, and TMEM119, together with CD31 labeling for ECs in D40 VORs (*Figure 2F*). Interestingly, DLL4 and EPHB4, which mark the venous and arterial ECs, respectively (*Vanlandewijck et al., 2018*; *Zhao et al., 2018*), were found to express only in separate EC populations (*Figure 2G and H*). This result indicates that ECs in VORs already underwent spontaneous functional maturation. Immunostaining also confirmed the presence of the venous and arterial EC subtypes (*Figure 2—figure supplement 1D and E*). Thus, the formed VORs contained the repertoire of brain vessel cell types resembling that in vivo.

To depict the developmental process of VORs, we reconstructed the time course of vascular cell developmental trajectory in pseudo-time (*Figure 3A*, *Figure 3—figure supplement 1A*). Five developmental stages and two time points were showed in the trajectory, with stages 1 and 2 representing

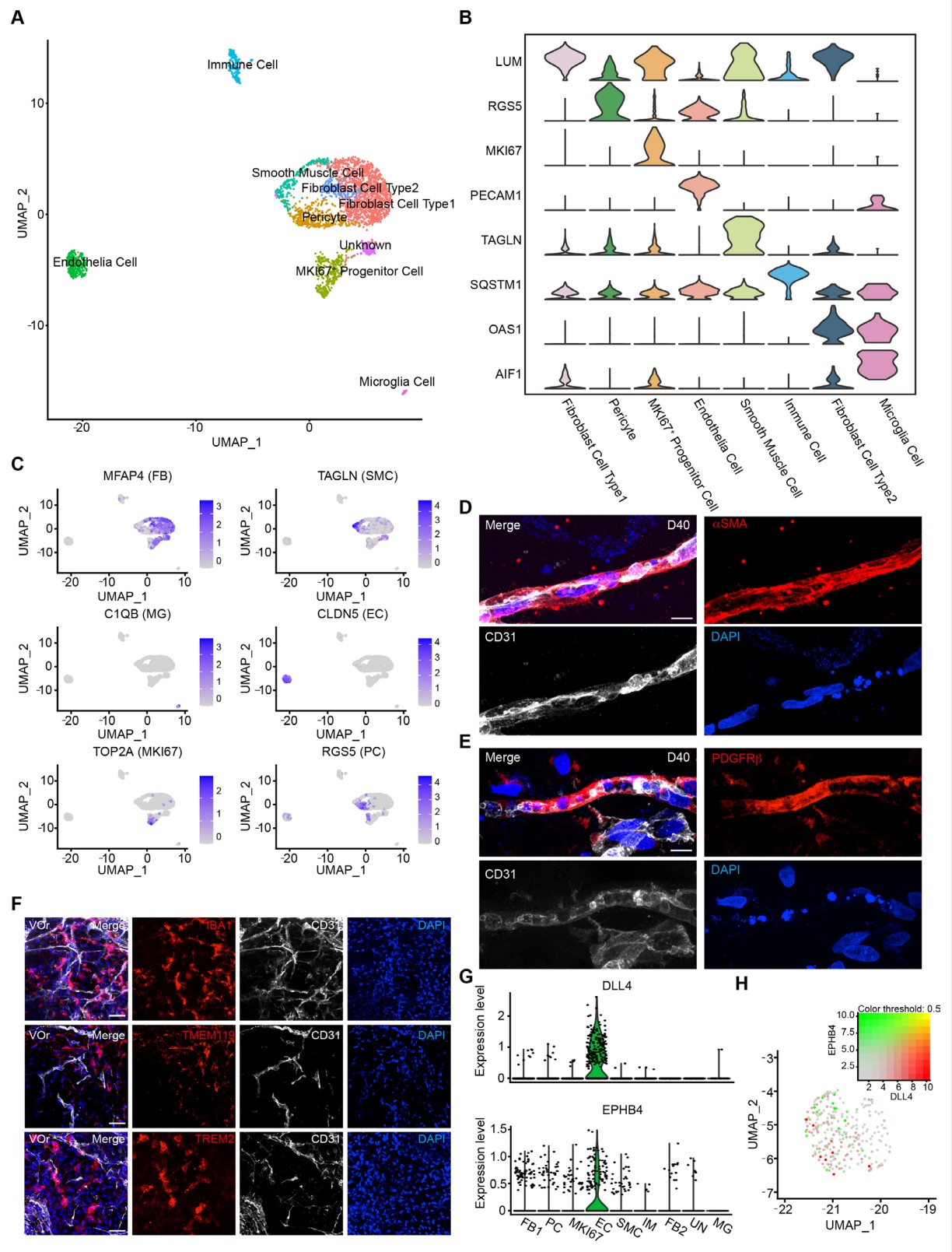

**Figure 2.** Single-cell transcriptomic analysis of vessel organoids (VOrs). (**A**) UMAP plot showing the nine major cell types isolated from day (D) 40 VOrs. (**B**) Violin plots showing the expression value of the typical markers in each cluster. (**C**) Expression pattern of cell-type-specific markers in VOrs. Relative expression level is plotted from gray (low) to blue (high) colors. (**D**) Immunostaining of αSMA for representing the smooth muscle cells in VOrs. Scale bar, 10 µm. (**E**) Immunostaining of PDGFRβ for representing the pericytes in VOrs. Scale bar, 10 µm. (**F**) Immunostaining of microglia markers (IBA1,

*Figure 2 continued on next page*

*Figure 2 continued*

TREM2, TMEM119) and endothelial marker CD31 in VORs at D40. Scale bar, 20 μm. (**G**) Violin plots showing the expression value of the venous marker EPHB4 and arterial marker DLL4 in endothelial cell (EC) clusters. (**H**) Expression pattern of arterial and venous markers in EC clusters. Relative expression level is plotted from gray to green (EPHB4) or red (DLL4) colors.

The online version of this article includes the following figure supplement(s) for figure 2:

**Figure supplement 1.** Cell-type analysis for vessel organoids (VORs) by scRNA-seq and immunostaining.

initial states, stage 3 representing the intermediate state, and stages 4 and 5 the latest (*Figure 3—figure supplement 1A and B*). Then, we used a panel of markers to annotate the main cell types and found that FB and PC were among the early developed cell types while the EC and MG were among the later ones (*Figure 3B–D*, *Figure 3—figure supplement 1C*). It is known that PC and SMC constitute mural cells of blood vessels and it has been difficult to distinguish them because they have similar gene expression profiles (*Smyth et al., 2018*). Using the developmental trajectory analysis, we found that PC appeared earlier than SMC (*Figure 3E*). PC markers (MEF2C, PEGFRβ, RGS5) were highly expressed in the early stages but downregulated in the later stages, while SMC markers (ACTA2, MYL9, TAGLN) showed opposite tendency (*Figure 3E*).

In order to determine to what extent the VORs resembled the brain vessels in vivo, we analyzed two accessible datasets for comparison. First, we compared the VORs and mouse cerebrovascular scRNA-seq data (*He et al., 2018*; *Vanlandewijck et al., 2018*) and found that the molecular features of the five major cell types, including FB, SMC, PC, EC, and MG, in VORs were similar to the counterparts of mouse cerebrovascular system (*Figure 3F*). We then referred to a dataset of scRNA-seq from eight adults and four embryonic human cortexes, which clustered a small number of vascular cell types, including EC, PC, and MG (*Polioudakis et al., 2019*). We analyzed the correlation of these three clusters in VORs, mouse, and human samples together and found that VORs and human showed stronger correlation in EC and PC clusters, while MG showed the highest consistency across all three datasets (*Figure 3G*). This result further confirmed the presence of brain-specific MG cells in VORs culture system with the introduction of neurotrophic factors. Thus, VOr is an appropriate model for the analysis of human cerebrovascular development in vitro.

Next we analyzed differentially expressed genes (DEGs) in ECs between VORs and mouse, human, and mouse, respectively, and found that a big fraction of genes upregulated were overlapped between the two sets of comparisons (*Figure 3H*). Remarkably, most of the top DEGs between VORs and mouse groups showed similar tendency in human samples (*Figure 3I*), suggesting the high similarity of ECs in VORs compared to that in human samples in vivo. The Gene Ontology (GO) analysis showed that the shared DEGs between VOr vs. mouse and human vs. mouse pairs were related to the angiogenesis pathway, suggesting that human vascular development may be more complex and robust than that of mouse (*Figure 3J*). We also analyzed DEGs within PCs and found that the majority of top changed genes in VORs compared with mouse samples were also present in DEGs of human vs. mouse pair comparison (*Figure 3—figure supplement 1D and E*). To further validate that VORs can faithfully mimic the process of vascular development in vivo, the expression of marker genes of three major cell types from VORs (EC, PC, MG) were compared with that of human samples. As shown in *Figure 3K*, marker genes of VORs were also highly expressed in the same cell types of human samples, further indicating the similarity of corresponding cell type. Recently, Lu et al. demonstrated that some in vitro-induced brain vessel cells lacked functional attributes of ECs but were more related to the neuroectodermal epithelial lineage-induced brain microvascular endothelial cells (Epi-iBMEC) (*Lu et al., 2021*). We performed principal component analysis (PCA) for ECs in VORs and other 28 datasets from this study, including the primary ECs, induced ECs (iEC), and Epi-iBMECs, and found that ECs in VORs showed a clear disparity from Epi-iBMECs but higher similarity to EC lineage (*Figure 3—figure supplement 1F*). The top and bottom loading genes showed separate endothelial and epithelial cell-type identities in these datasets (*Figure 3—figure supplement 1G*) and VORs exhibited strong EC properties (*Figure 3—figure supplement 1G*). These results indicate that the VOr model can be used to recapitulate cerebrovascular development in vitro.

## Generation of fusion vascularized brain organoids

Having established the VORs, we decided to generate vascularized BOrs by using co-culture strategy. For this purpose, we established the induction system of BOrs from human H9 ESCs according to the

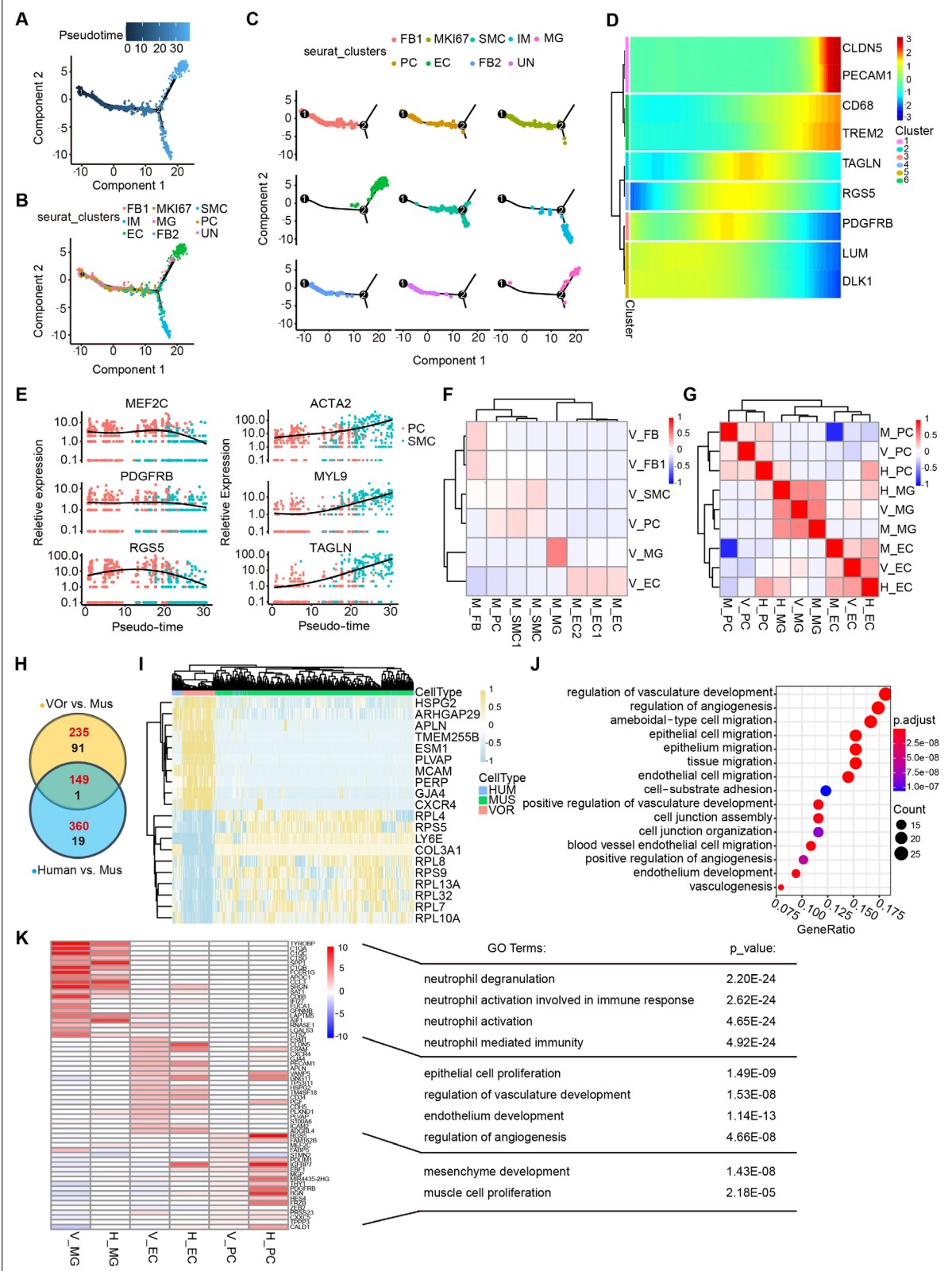

**Figure 3.** Cell fate trajectory analysis in vessel organoids (VOrs) and the comparison with cell types in vivo. (**A**) Single-cell trajectories by monocle analysis showing developmental stage of the VOrs. (**B**) Clusters in UMAP showing trajectory track. (**C**) Developmental trajectory of indicated cell clusters in VOrs. (**D**) Heatmap showing the expression level of the main cell type-specific markers with pseudo-time. (**E**) Expression of markers in smooth muscle cell (SMC) and pericyte (PC) with pseudo-time. (**F**) Correlation analysis of cell clusters (endothelial cell [EC], microglia [MG], PC, SMC, fibroblast [FB])

*Figure 3 continued on next page*

*Figure 3 continued*

between VOrs and mouse brain. V, data from VOrs; M, data from mouse. (**G**) Correlation analysis of cell clusters (EC, MG, PC) among VOrs, mouse, and human brain single-cell data. V, data from VOrs; M, data from mouse; H, data from human. (**H**) Venn diagram showing the differentially expressed genes (DEGs) in EC clusters for VOr and human samples compared to mouse samples. Red for upregulated genes, black for downregulated genes. (**I**) Heatmap showing the top enriched DEGs in the EC cluster for VOrs samples compared to mouse sample (fold change > 1.25 and p<0.05). (**J**) Gene Ontology (GO) analysis of the 149 upregulated DEGs in (**H**) (p-value<0.1 and false discovery rate [FDR]<0.05). (**K**) Top 20 marker genes for VOrs in the main clusters (EC, PC, MG) (fold change > 1.25 and p<0.05) compared to human sample, with significant pathways by GO analysis (p-value<0.1 and FDR < 0.05). V, data from VOrs; H, data from human.

The online version of this article includes the following figure supplement(s) for figure 3:

**Figure supplement 1.** Cell types in vessel organoids (VOrs) are similar to that of human samples in vivo.

---

methods reported previously (*Lancaster and Knoblich, 2014*; *Mariani et al., 2012*; *Ou et al., 2020*; *Hou et al., 2021*), with some modifications (*Figure 4—figure supplement 1A*). Cerebral organoids at different developmental stages were stained with neural progenitor markers PAX6 and phospho-vimentin (p-VIM), the proliferation marker KI67, intermediate progenitor marker TBR2, young neuron marker DCX (doublecortin), mature neuron marker TUJ1, and the cortical layer markers (TBR1, CTIP2, SATB2, REELIN), and the results indicated that the BOrs were well induced (*Figure 4—figure supplement 1B–E*). As expected, CD31$^+$ ECs were barely seen in this induction system (*Figure 4—figure supplement 1F*). After the step of neural ectoderm induction, EBs with neuroepithelial (NE) property were co-embedded with VPs in one Matrigel droplet, and then cultured under the condition of VOrs maturation with the medium containing N2 and B27 (*Figure 4A*). For better invasion of vessels into the developing BOrs, we put two VP bodies in both sides of one NE body (*Figure 4A*). After co-culture for different days, VOrs labeled by GFP gradually wrapped BOrs and finally formed a fused vasculature and brain organoids (fVBOrs) by D40 (*Figure 4B*). Whole-mount staining of the fVBOrs showed that DCX-labeled neurons were enwrapped by invaded vessels labeled by CD31 (*Figure 4C*). The fVBOrs were positively labeled by Human-Nuclei (HUNU), indicating human cell identity (*Figure 4—figure supplement 2A*).

Supported by pericytes and astrocytes, the brain microvascular ECs form a particularly tight layer called the blood–brain barrier (BBB), which selectively controls the flow of substances into and out of the brain by forming complex intercellular tight junctions and protects the brain from harmful substances (*Augustin and Koh, 2017*; *Chow and Gu, 2015*; *Lippmann et al., 2012*; *Sweeney et al., 2019*). To determine whether the fVBOrs developed BBB-like features, we examined the expression of the tight junction proteins Claudin5 (CLDN5) and ZO-1 (*Figure 4D*, *Figure 4—figure supplement 2B and C*), and the efflux transporter p-glycoprotein, which helps the recycling of small lipophilic molecules diffused into ECs back to the blood stream (*Augustin and Koh, 2017*; *Lippmann et al., 2012*; *Figure 4—figure supplement 2D*). Notably, stronger CLDN5 signals were observed in BOr regions in contact with vessels, suggesting the appearance of tight junctions-like structures (*Figure 4—figure supplement 2B*). Furthermore, astrocyte-like cells labeled by S100 or GFAP were also observed in fusion organoids, forming neurovascular unit-like structures composed of CD31/GFP-labeled vascular structures and MAP2-labeled neurons (*Figure 4E*). The presence of neurovascular unit structure in fusion organoids was also confirmed by transmission electronic microscopy (TEM), which showed the EC basement membrane enclosed by pericytes and tightly contacted by end feet of astrocytes (*Figure 4—figure supplement 2E*).

Next, we examined the functionality of BBB in the fVBOrs by measuring the permeability of molecules with different BBB penetration capability (*Bergmann et al., 2018*; *Cho et al., 2017*; *Dai et al., 2018*; *Xu et al., 2019*). The selectivity of BBB was determined by incubating fVBOrs with rhodamine-labeled Angiopep-2, a peptide capable of permeating through BBB selectively (*Bergmann et al., 2018*; *Cho et al., 2017*). We found that Angiopep-2 exhibited strong signals in the fVBOrs, but scrambled peptides displayed no detectable signal (*Figure 4F and G*). The z-stack images showed that the intensity of Angiopep-2 signals decreased from the surface to the inner of fVBOrs (*Figure 4F*, bottom). In contrast to fusion organoids, the BOrs alone showed much weaker Angiopep-2 signals (*Figure 4—figure supplement 2F and G*). Taken together, these results indicate that fVBOrs have developed BBB structures with selective permeability.

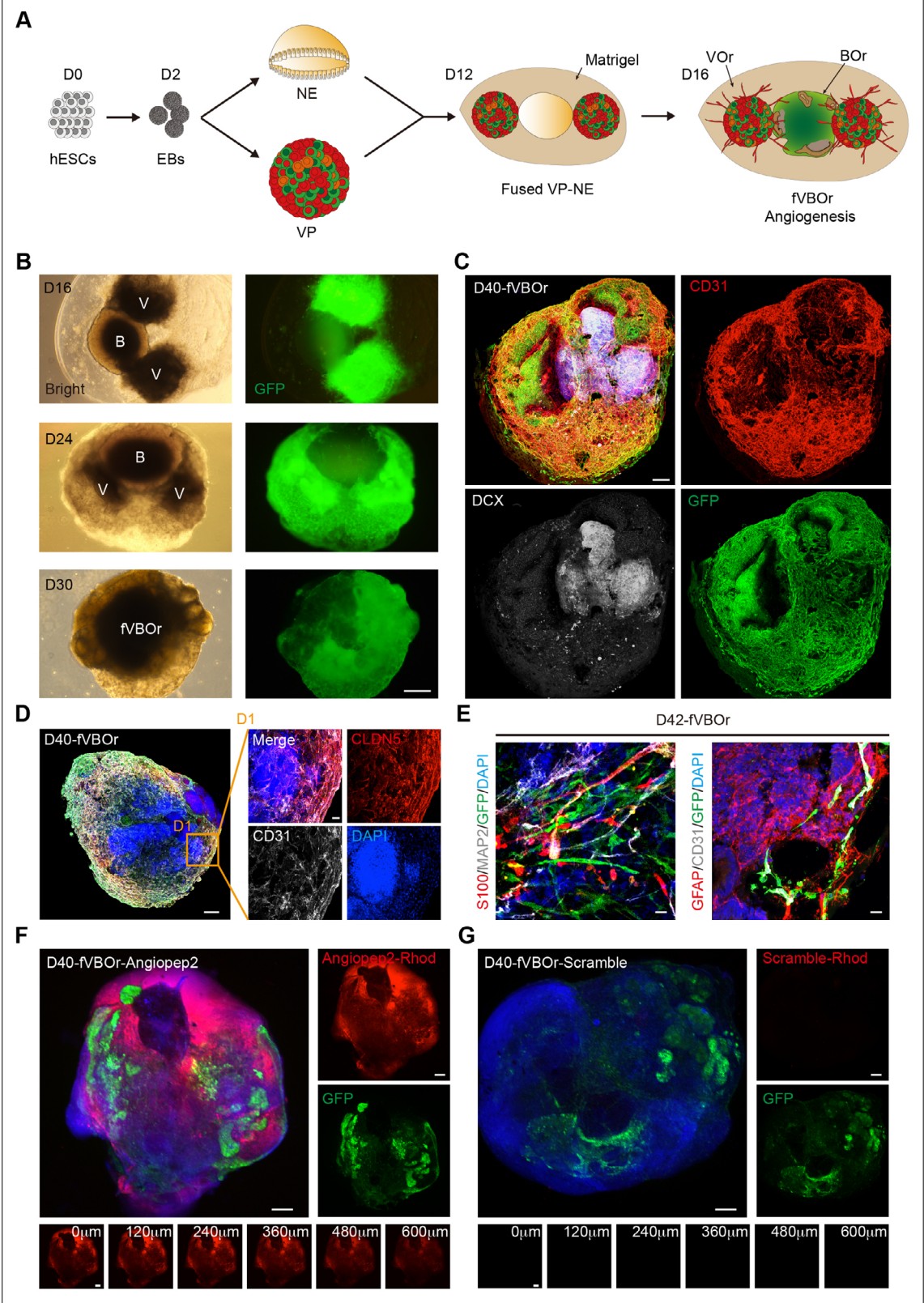

**Figure 4.** Generation of fused vasculature and brain organoids (fVBOrs) with blood–brain barrier (BBB) structure. (**A**) Schematic view of the method for generating fVBOrs. EBs, embryonic bodies; NE, neuroepithelium; VP, vascular progenitor; VO, vessel organoid; BOr, brain organoid. (**B**) fVBOrs at different developmental stages. Scale bar, 500 μm. V, VOr; B, BOr. (**C**) Immunostaining of CD31 and DCX for labeling vessels and neurons, respectively, in day (D) 40 fVBOrs. Scale bar, 200 μm. (**D**) Immunostaining of CLDN5 for labeling tight junctions in fVBOrs. Scale bar, 200 μm. **D1**, enlarged area.

*Figure 4 continued on next page*

*Figure 4 continued*

(**E**) Immunostaining for markers of astrocytes (S100/GFAP), neurons (MAP2), endothelial cells (CD31), and vessel structures (GFP) in fVBORs. Orange arrows indicate astrocytes end feet. Scale bar, 20 μm. (**F, G**) Confocal fluorescence images showing the transport of rhodamine-labeled angiopep-2 (Angiopep-2–Rhod), rhodamine–scramble peptide (Scramble–Rhod) in fVBORs. Scale bar, 200 μm. Bottom, z-stack images of rhodamine signals.

The online version of this article includes the following figure supplement(s) for figure 4:

**Figure supplement 1.** Generation of human brain organoids (BORs).

**Figure supplement 2.** Blood–brain barrier (BBB)-like structures in fused vasculature and brain organoids (fVBORs).

## Microglia cells in fVBORs are responsive to immune stimuli and can engulf synapses

It is generally believed that MGs are developed from the yolk-sac progenitors, which then populate in the developing brain to regulate neurogenesis and neural circuit refinement (*Kaur et al., 2017*; *Mosser et al., 2017*; *Salter and Stevens, 2017*). Indeed, in the unguided cerebral organoids, spontaneous MG can emerge (*Ormel et al., 2018*), probably due to the presence of residue mesodermal progenitors (*Quadrato et al., 2017*). However, the functional investigation is limited due to the variable and inconsistent batch effects. Based on the scRNA-seq and staining results suggesting the presence of MG in VORs, we decided to explore the possibility of introducing these MG-like cells into BORs using fusion strategy. To this end, we first determined the molecular features of MG in VORs. As shown in *Figure 5A*, the MG identity was confirmed by the expression of specific marker genes *AIF1* (the gene encoding IBA1) and *CD68*. The GO analysis showed that the functions of MG markers were mainly concentrated on the pathways of immune and inflammatory responses (*Figure 5B*). During the VOr culture, the expression of MG marker genes, such as *AIF1* or *TMEM119*, gradually increased (*Figure 5C*), indicating again the MG induction. In line with this notion, D40 VORs exhibited increased abundance of MG with amoeboid-like morphology compared to D25 MG mostly with round morphology (*Figure 5D and E*).

We next examined whether MG could migrate from VORs into BORs after fusion. We found large amount of IBA1$^+$GFP$^+$ MG-like cells in the neural part of fVBORs, whereas BORs alone had no MG-like signal (*Figure 5F*). Thus, fVBORs also contained MG-like cells, besides vasculatures. Notably, there were a number of IBA1$^+$GFP$^-$ cells in fVBORs (*Figure 5F*). Based on the brain RNA-seq database of various cell types in mouse and human (*Zhang et al., 2014*), VEGFR1 (encoded by *FLT1* gene) is highly expressed in human microglia (data not shown). It remains possible that the residue microglia in BORs sense the signals, such as VEGF, from the invading vessels to be motivated and activated. Next, we determined the responsiveness of these MG-like cells to the treatments used to diminish or activate microglia to verify the cell identity. First, we treated D30 VORs with PLX5622, a selective inhibitor of colony-stimulating factor 1 receptor (CSF1R), which was used to ablate MG in mice (*Huang et al., 2018*). After treatment for 7 days with 2 μM PLX5622, the IBA1-labeled cells were almost completely gone, and the ablation effect lasted for at least 3 days in the absence of PLX5622 (*Figure 5—figure supplement 1A and B*). Likewise, the PLX5622 treatment also depleted MG-like cells in D40 fVBORs (*Figure 5—figure supplement 1C*). These results further confirmed the identity of MG-like cells in VORs and fVBORs. Next, fVBORs were treated with 0.5 μg/ml lipopolysaccharide (LPS) for 72 hr to induce inflammatory response. The LPS stimulation caused marked increase in the expression of inflammatory factors TNFα and IL-6 (*Figure 5G*). Interestingly, the expression levels of TNFα or IL-6 were attenuated in PLX5622-treated fusion organoids (*Figure 5G*), suggesting the involvement of MG-like cells in LPS-induced immune response. These results support the conclusion that MG-like cells possess responsive ability to immune stimuli.

MGs also play important roles in synapse elimination by engulfing synapses and promote the process of neuronal maturation (*Eroglu and Barres, 2010*; *Filipello et al., 2018*; *Gunner et al., 2019*; *Popova et al., 2021*; *Schafer et al., 2012*; *Scott-Hewitt et al., 2020*; *Zuchero and Barres, 2015*). To further strengthen the conclusion that MGs in fVBORs were functional, we examined the ability of MGs in synaptic engulfment by double staining of postsynaptic density protein 95 (PSD95) and MG marker IBA1. Remarkably, many PSD95-labeled puncta were distributed within MG-like cells, indicating synaptic engulfment (*Figure 5H*). Next, we performed the electrophysiological recoding of neurons in BORs and fVBORs to measure neuronal activity. We found that the frequency of spontaneous excitatory post-synaptic currents (sEPSCs) in fVBORs significantly decreased, while the amplitude of sEPSC

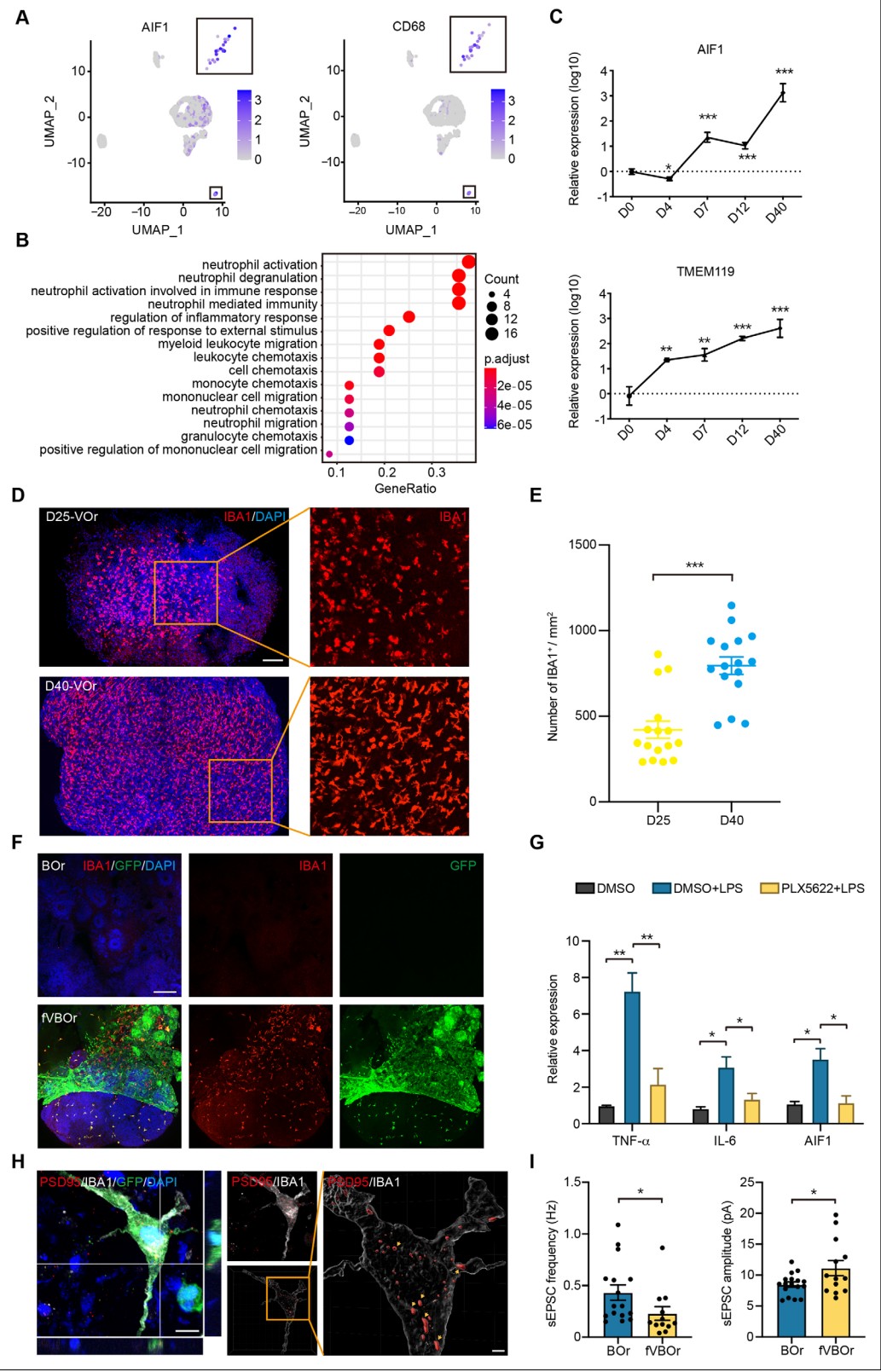

**Figure 5.** Microglial cells in fused vasculature and brain organoids (fVBOrs). (**A**) UMAP plot showing single-cell expression pattern of microglial-specific markers in vessel organoids (VOr). Relative expression level is plotted from gray to blue colors. (**B**) Gene Ontology (GO) analysis of microglial cell marker genes (p-value<0.1 and false discovery rate [FDR] < 0.05). (**C**) qPCR analysis for expression of microglial markers AIF1 and TMEM119 in

*Figure 5 continued on next page*

*Figure 5 continued*

developing VOrs. Data are presented as mean ± SEM (n = 3 independent experiments with 6–7 organoids in each group at indicated time point). Error bars indicate SEM. **p<0.01, ***p<0.001. (**D**) Immunostaining of IBA1 for labeling microglial cells in day (D) 25 and D40 VOrs. Scale bar, 200 µm. (**E**) Quantification of the IBA1⁺ cell number in D25 and D40 VOrs. n = 16. Error bars indicate SEM. Student's *t*-test, ***p<0.001. (**F**) Immunostaining of IBA1 for labeling microglial cells in BOrs and fVBOrs, respectively. Scale bar, 200 µm. (**G**) qPCR analysis for the expression of indicated genes in D40 fVBOrs treated with lipopolysaccharide (LPS) (500 ng/ml, MCE, HY-D1056) without or with PLX5622 2 µM (MCE, HY-11415) using DMSO as vehicle control. Relative expression was normalized to GAPDH. n = 3 independent experiments with 8–10 organoids in each group. Error bars indicate SEM. One-way ANOVA, *p<0.05, **p<0.01. (**H**) Double immunostaining and orthogonal view of IBA1 and PSD95 signals within microglia (MG)-like cells (left) and 3D-surface-reconstructed image (right). Arrows indicate synapse puncta engulfed in MG-like cells. Scale bar, 10 µm (left); 2 µm (right). (**I**) Quantification of the spontaneous excitatory post-synaptic current (sEPSC) frequency (left) and amplitude (right) in neurons of D70 BOrs and fVBOrs. Data are presented as mean ± SEM (BOrs: n = 16 neurons from six organoids; fVBOrs: n = 13 neurons from six organoids). Error bars indicate SEM. Two-tailed Student's *t*-test. *p<0.05.

The online version of this article includes the following figure supplement(s) for figure 5:

**Figure supplement 1.** PLX5622 ablates microglias (MGs) in vessel organoids (VOrs).

**Figure supplement 2.** Whole-cell patch-clamp recoding of neurons in brain organoids (BOrs) and fused vasculature and brain organoids (fVBOrs).

---

markedly increased (*Figure 5I*, *Figure 5—figure supplement 2A and B*). These results are in line with the role of MGs in regulating synaptic refinement. The neurons in fVBOrs also exhibited increased inward current amplitudes when clamped at 50 and 60 mV voltage, indicating the promoted neuronal excitability in fVBOrs (*Figure 5—figure supplement 2C*). Thus, neurons in fVBOrs exhibited accelerated functional maturation process, largely owing to the integration of vasculature and/or MG-like cells.

## Increased neural progenitors in the fusion organoids

It has been shown that vasculature acts as a critical niche that helps maintain the survival and stemness of neural progenitors, and EC-derived soluble factors might contribute to this function (*Delgado et al., 2014*; *Ottone et al., 2014*; *Shen et al., 2004*). Prompted by this information, we compared neurogenesis patterns in fVBOr and BOr. Interestingly, the fVBOrs exhibited marked increase in the thickness of NE rosettes compared to BOrs at the same corresponding stages (D25) (*Figure 6A and B*). In line with this notion, the density of neural progenitors (NP) marked by PAX6 or mitotic NPs marked by p-VIM also increased in fusion organoids (*Figure 6C–E*). However, the density of DCX-labeled differentiated neurons or TBR1-labeled early-born cortical neurons had no difference during the observation period (*Figure 6F and G*, *Figure 6—figure supplement 1A and B*). These results suggest that the factors produced by VOrs might promote the proliferation of NPs after fusion with BOrs, with little effect on neuronal differentiation. In the classical BOr culture system, the inner cells are extremely vulnerable to limited accessibility to the trophic factors in culture medium. In line with this notion, BOrs at D40 or D70 showed abundant apoptotic cells expressing cleaved caspase 3 (c-CASP3) in the central regions, whereas the apoptotic cells were markedly reduced in fVBOrs (*Figure 6—figure supplement 1C–E*). This result indicates that the vessels in fVBOrs may protect neural cells from cell death. The diffusion of oxygen by vascular structures or some protective factors secreted by vascular cells may contribute to this role. Thus, the fusion organoids generated in this work can be used to study interactions among multiple cell types during brain development.

## Discussion

The prevalent approaches for BOr induction start from neuroectoderm induction using BMP and/or Wnt signaling inhibitors, or neural induction medium to treat embryoid bodies, followed by brain regional differentiation, and neural maturation (*Giandomenico and Lancaster, 2017*; *Kadoshima et al., 2013*; *Lancaster and Knoblich, 2014*; *Lancaster et al., 2013*; *Qian et al., 2016*). Although this system is powerful, it lacks vascularization, rendering it impossible to recapitulate vascular–neural interactions and model-related diseases. Here, we developed an approach for generating neural-specific VOrs by

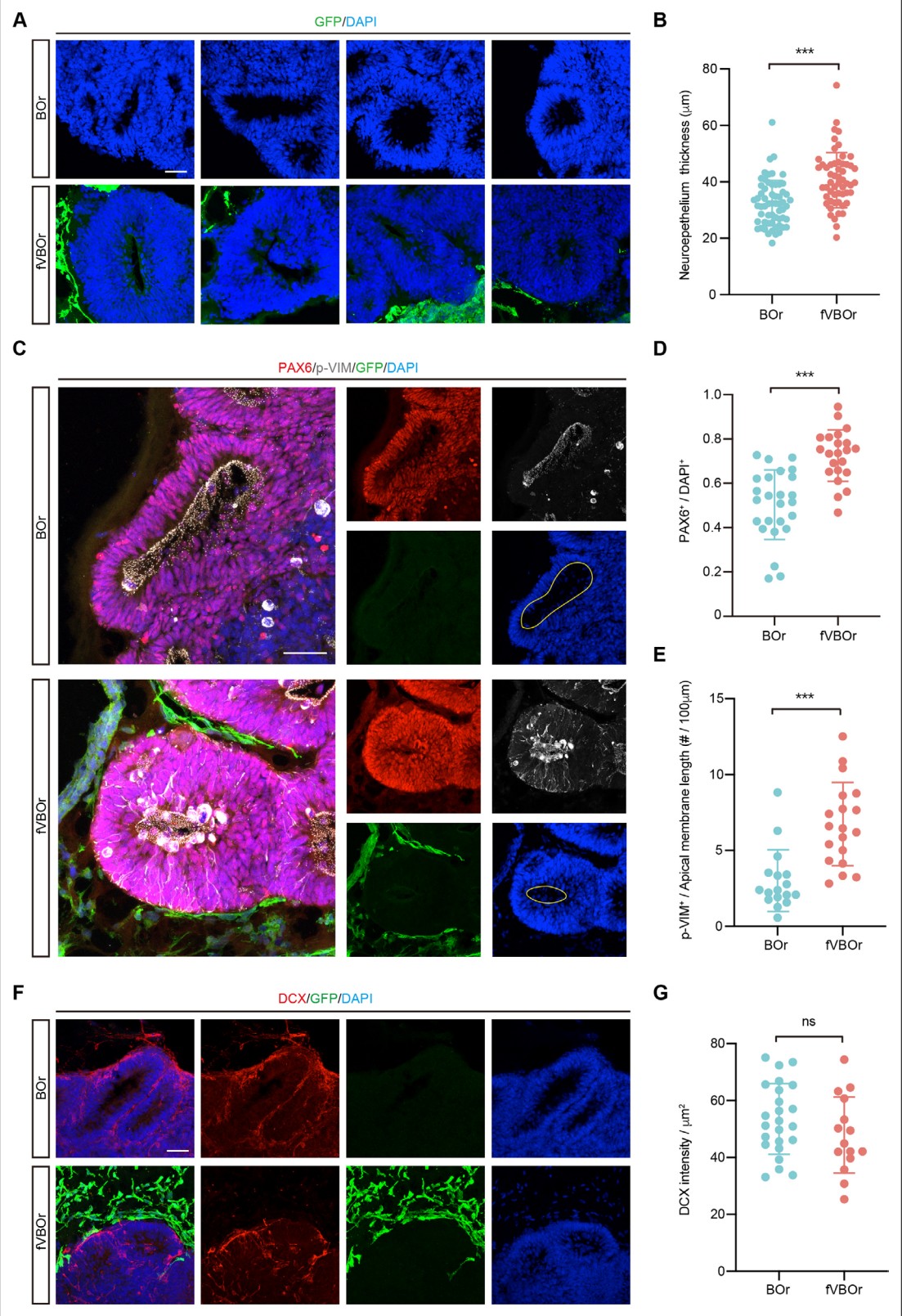

**Figure 6.** Increased neurogenesis in fused vasculature and brain organoids (fVBOrs). (**A**) Immunostaining for DAPI showing the neuroepithelium rosettes of brain organoids (BOrs) and fVBOrs at D25. Scale bar, 50 µm. (**B**) Quantification of neuroepithelium thickness of BOrs and fVBOrs. Data are presented as mean ± SEM (BOrs: n = 60 rosettes from seven organoids; fVBOrs: n = 55 rosettes from six organoids). Error bars indicate SEM. Two-tailed Student's *t*-test. ***p<0.001. (**C**) Immunostaining for PAX6 and phospho-vimentin (p-VIM) in VZ-like area of BOrs and fVBOrs at day (D) 25. Scale bar, 50 µm. Apical

*Figure 6 continued on next page*

*Figure 6 continued*

membrane is shown in yellow circle. (**D, E**) Quantification of the density of PAX6$^+$ (**D**) and the density of p-VIM$^+$ cells per 100 µM apical membrane length (**E**) in BOrs and fVBOrs. Data are presented as mean ± SEM (PAX6: n = 25 rosettes from four organoids; p-VIM: n = 23 rosettes from four organoids). Error bars indicate SEM. Two-tailed Student's *t*-test. ***p<0.001. (**F**) Immunostaining for DCX in BOrs and fVBOrs at D25. Scale bar, 50 µm. (**G**) Quantification of the intensity of DCX in BOrs and fVBOrs. Data are presented as mean ± SEM (BOrs: n = 24 rosettes from three organoids, fVBOrs: n = 15 rosettes from four organoids). Error bars indicate SEM. ns, no significant difference (p=0.308, two-tailed Student's *t*-test).

The online version of this article includes the following figure supplement(s) for figure 6:

**Figure supplement 1.** Reduced apoptotic areas in fused vasculature and brain organoids (fVBOrs).

initial transient mesoderm induction, sequential VP and EC induction, followed by treatments with neurotrophic reagents. Then, we established an integrated vascularized BOr model by fusing the BOrs and VOrs. This vascularization strategy considered the compatibility to mesodermal and ectodermal linages and led to generation of BOrs with complex tubular vessels, functional neurovasculature units, as well as microglia responding to immune stimulation.

The fVBOr model generated in this study provides a possibility to analyze the process of brain angiogenesis and complex interactions between vasculatures and neural cells. It has been shown previously that brain vascularization is regulated by neural progenitors (*Matsuoka et al., 2017*), and, on the other hand, vasculatures promote neurogenesis and oligodendrocyte precursor migration (*Tata et al., 2016*; *Tsai et al., 2016*). Notably, we found that in fVBOrs only vasculatures close to or located in the BOrs expressed tight junction markers, including CLDN5 and ZO-1 (*Figure 4—figure supplement 2B and C*), consistent with the idea that BBB maturation is regulated by neural cues (*Lippmann et al., 2012*). The effects of vasculatures on neurogenesis were also found in fVBOrs, which exhibited increased pool of neural progenitors and reduced apoptosis (*Figure 6C–E*, *Figure 6—figure supplement 1C–E*). The reduction in apoptotic cells was also seen in grafted BOrs with the invasion of host blood vessels (*Mansour et al., 2018*; *Shi et al., 2020*). It is conceivable that the vascularized BOrs developed in this study may provide a feasible platform for the study of human brain development, vasculature-related diseases, or pharmaceutical interventions, which need to pass the BBB.

Several studies have tried to generate BBB-like structures in vitro by culturing PSC-derived ECs (*Lippmann et al., 2012*; *Qian et al., 2017*) or co-culturing PSC-derived cells without or with primary cells in 2D system (*Appelt-Menzel et al., 2017*; *Canfield et al., 2017*) or 3D system (*Bergmann et al., 2018*; *Cho et al., 2017*). Although some features of the BBB were reproduced in these studies, the tube-like structures of blood vessels were lacking. The vessel organoid model we have established showed intense vascular network with characteristics and specificity of human cerebral vessels based on transcriptomic analysis and functional assay. It has been demonstrated that the combination of PSC-derived tissue-specific progenitors or relevant tissue samples with ECs and mesenchymal stem cells can generate vascularized organs (*Takebe et al., 2015*; *Takebe et al., 2013*), but different derivation of vascular ECs and the brain from distinct germ layers limits the vascularization in BOrs initialized from neuroectoderm induction. Indeed, we failed to observe any vascular structures in the BOrs generated using the prevalent approach (*Figure 4—figure supplement 1F*). Although the fVBOr model has shown branched vessels, it still lacks active blood flow. One possible approach to solve the problem is to combine microfluidic techniques and organoid cultures and make an 'organ-on-a-chip,' which may authentically mimic the vascular environment. In further study, the functional perfusion of vessels in fVBOrs can be verified via in vivo grafting that allows integration of vessels between organoids and the hosts as that done in other studies (*Pham et al., 2018*; *Cakir et al., 2019*; *Cakir et al., 2019*; *Shi et al., 2020*).

The blood vessels in the brain are not only required for oxygen and nutrient supply, but also involved in regulating neurogenesis. The fusion organoids allow for recapitulating early developmental processes like vasculogenesis and angiogenesis, as well as the integration of microglia. Unlike BBB-like structures generated in other studies that combined various mature cell types (*Cho et al., 2017*; *Lippmann et al., 2012*; *Qian et al., 2017*), the ones generated in our study were directly induced from PSCs, which resembled the developmental processes in vivo. Although the factors that defined the identity of microglia cells in VOrs are not clear, their migration into the BOrs resembled the extra-embryonic originality. It is known that astrocytes are essential components of the neurovascular units (*Abbott et al., 2006*), and their involvement in the maturation of BBB-like units or immune surveillance awaits further investigation.

## Materials and methods

### Cell lines

H9 human embryonic stem (H9-hES) cell line was purchased from iMedCell, the identity of which was confirmed by STR profiling (performed by Applied Cell). The cells were tested for mycoplasma and the result was negative. H9-hES-EGFP was generated by introducing the CAG-EGFP DNA fragment into the genome locus ROSAβgeo26 (ROSA26) using the CRISPR/Cas9 method.

### hESCs culture

Both H9-hES and H9-hES-EGFP cells were cultured and passaged as previously described (*Ou et al., 2020*; *Ou et al., 2021*). Cells were cultured on hESC-Matrigel (Corning)-coated dishes in mTeSR1 (STEMCELL) medium with the addition of bFGF (4 ng/ml, STEMCELL). The culture medium was half-replaced every day, and then cells were passaged every 5 days using passage reagent ReLeSR (STEMCELL). Detailed information about recombinant proteins and chemical compounds is available in Appendix 1—key resources table.

### Generation of human brain organoid

hESC clones were dissociated into single cells with Accutase (STEMCELL), then cells were resuspended in mTeSR1 medium containing 10 μM Y27632 (STEMCELL), and seeded into the lipidure-coated (NOF CORPRATION) V-bottom 96-well plate (Thermo) with 7000 cells per aggregate, 150 μl per well to form EBs. On day 2, the culture medium was replaced by the ectodermal induction medium (DMEM/F12 [Life/Invitrogen] containing 20% [v/v] Knockout Serum Replacer [Gibco], 1% [v/v] MEM-NEAA [Gibco], 3.5 μl/l β-mercaptoethanol [Sigma-Aldrich], 1% [v/v] GlutaMAX [Gibco], 2.5 μM dorsomorphine [Tocris], and 2 μM A83-01 [Tocris]). On day 4, the ectodermal induction medium was half-replaced. On day 6, the EB medium was replaced by the neural induction medium (DMEM/F12 containing 1% [v/v] N2 supplement [Life/Invitrogen], 1% [v/v] MEM-NEAA, 1% [v/v] GlutaMAX, 1 μg/ml heparin [Sigma-Aldrich], 10 μM SB431542 [Selleck], and 200 nM LDN193189 2HCL [Selleck]) and lasted for 6 days. The neural induction medium was half-renewed every other day. On day 12, the EBs were embedded into growth factor-reduced Matrigel droplet (Corning), as described previously (*Lancaster and Knoblich, 2014*). The culture medium was replaced by the differentiation medium (50% [v/v] DMEM/F12 and 50% [v/v] Neurobasal medium [Life/Invitrogen] containing 0.5% [v/v] N2 supplement, 0.5% [v/v] B27 supplement without vitamin A [Life/Invitrogen], 3.5 μl/l β-mercaptoethanol, 250 μl/l insulin [Sigma-Aldrich], 1% [v/v] GlutaMAX, and 0.5% [v/v] MEM-NEAA, 1% [v/v] Antibiotic-Antimycotic [Gibco]). After 4 days, the differentiation medium was replaced by the maturation medium (50% [v/v] DMEM/F12 and 50% [v/v] Neurobasal medium containing 0.5% [v/v] N2 supplement, 0.5% [v/v] B27 supplement [Life/Invitrogen], 3.5 μl/l β-mercaptoethanol, 250 μl/l insulin, 1% [v/v] GlutaMAX, and 0.5% [v/v] MEM-NEAA, 1% [v/v] Antibiotic-Antimycotic). Then, the organoids were transferred into a shaker in the 5% $CO_2$ incubator at 37°C for maturation, and the medium was renewed every 3–4 days. Detailed information about recombinant proteins and chemical compounds is available in Appendix 1—key resources table.

### Generation of human vessel organoid

H9-hES-GFP clones were dissociated into single cells with Accutase, then the cells were resuspended in the mTeSR1 medium containing 10 μM Y27632 and seeded into the lipidure-coated V-bottom 96-well plate with 9000 cells per aggregate, 150 μl per well to form EBs. On day 2, the culture medium was replaced by the mesodermal induction medium (APEL2 [STEMCELL] with 6 μM CHIR99021 [Selleck]). On day 4, the mesodermal medium was replaced by endothelial induction medium (APEL2 with 50 ng/ml VEGF [STEMCELL], 25 ng/ml BMP4 [R&D, 314BP] and 10 ng/ml bFGF). On day 7, medium was changed into MV2 medium (PromoCell) with 50 ng/ml VEGF for the maturation of ECs, and the medium was renewed every other day. From day 12, the EBs were embedded into Matrigel droplets and cultured with VEGF-containing (20 ng/ml) neural differentiation medium, as that used for BOr culture. Detailed information about recombinant proteins and chemical compounds is available in Appendix 1—key resources table.

### Fusion of vascular brain organoids

To generate the fusion organoids, two VOr EBs and one BOr EB were collected and then embedded together into one Matrigel droplet (25 μl) on day 12. The two VOr EBs were put on two sides of the

BOr EB, and pipette tips could be used to adjust the shape and site of the three EBs. The following steps were the same as the nonfusion BOr EBs with the addition of 20 ng/ml VEGF.

## Immunofluorescence

The collected organoid samples were fixed in 4% paraformaldehyde (PFA) at 4°C overnight, and then washed three times with PBS, dehydrated in 30% sucrose at 4°C for 24–48 hr. Then, organoids were embedded in O.C.T (Sakura) and cryosectioned into 30-μm-thick slices. The sectioned slices were boiled in citrate-based antigen retrieval buffer for 10 min, followed by cooling for over 60 min. Slices were washed with PBS three times and incubated in 0.3% TritonX-100 (Sigma-Aldrich) at room temperature (RT) for 30 min, blocked with 5% BSA (Sigma-Aldrich) in 0.1% TritonX-100 at RT for 1 hr, incubated with the primary antibody at 4°C for over 48 hr, followed by washes with PBS and incubation with the secondary antibody at 4°C overnight. Secondary antibodies were Alexa Fluor 488, 555, 594, or 647-conjugated donkey anti-mouse, -rabbit, -rat, or -chicken IgG (Invitrogen, all used at 1:1000 dilution). DAPI (beyotime, 1:2000 dilution) was used to mark cell nuclei. Stained sections were mounted with mounting medium and stored at 4°C before imaging. All images were acquired by confocal imaging systems.

For whole-mount staining, the organoid samples were fixed in 4% PFA at 4°C overnight, and then washed three times with PBS, followed by the incubation in 0.5% TritonX-100 at RT for 1 hr. After blocking with 5% BSA in 0.1% TritonX-100 at RT for 1 hr, organoids were incubated with primary antibodies at 4°C for over 48 hr, washed with PBS, and then incubated with secondary antibodies at 4°C for over 48 hr. The stained organoids were washed with PBS three times before confocal imaging. Detailed information about primary antibodies is available in Appendix 1—key resources table.

## Quantitative PCR (qPCR)

The total RNA of 3–4 organoids was extracted using the RNeasy Plus Micro Kit (QIAGEN), followed by reverse transcription to generate cDNA with GoScript Reverse Transcription Kit (Promega). Quantitative PCR was performed by using the Agilent Mx3000P qPCR system with the 2×SYBR Green qPCR Master Mix (Bimake). Relative mRNA expression was determined by the delta cycle time with human GAPDH as the internal control in data normalization. Primer sequences were as follows:

> TNF-α: forward, 5′-CACAGTGAAGTGCTGGCAAC-3′, reverse, 5′-AGGAAGGCCTAAGGTCCACT-3′;
> IL-6: forward, 5′-TTCCAAAGATGTAGCCGCCC-3′, reverse, 5′-ACCAGGCAAGTCTCCTCATT-3′;
> GAPDH: forward, 5′-TCGGAGTCAACGGATTTGGT-3′, reverse, 5′-TTCCCGTTCTCAGCCTTGAC-3′;
> IBA1: forward, 5′-AAACCAGGGATTTACAGGGAGG-3′, reverse, 5′-GGGCAGATCCTCATCACTGC-3′;
> TMEM119: forward, 5′-GAGGAGGGACGGGAGGAG-3′, reverse, 5′-GACCAGTTCCTTGGCGTACA-3′;
> NANOG: forward, 5′-CAATGGTGTGACGCAGAAGG-3′, reverse, 5′-TGCACCAGGTCTGAGTGTTC-3′;
> OCT4: forward, 5′-CTCGAGAAGGATGTGGTCCG-3′, reverse, 5′-TGACGGAGACAGGGGGAAAG-3′;
> PECAM1: forward, 5′-AGACGTGCAGTACACGGAAG-3′, reverse, 5′-TTTCCACGGCATCAGGGAC-3′;
> VE-Cadherin: forward, 5′-CGCAATAGACAAGGACATAACAC-3′, reverse, 5′-GGTCAAACTGCCCATACTTG-3′;
> VWF: forward, 5′-CCCGAAAGGCCAGGTGTA-3′, reverse, 5′-AGCAAGCTTCCGGGGACT-3′;
> VEGFR2: forward, 5′-GAGGGGAACTGAAGACAGGC-3′, reverse, 5′-GGCCAAGAGGCTTACCTAGC-3′;
> VEGFR1: forward, 5′-AACGTGGTTAACCTGCTGGG-3′, reverse, 5′-AGTGCTGCATCCTTGTTGAGA-3′;
> PDGFR: forward, 5′-ATCAGCAGCAAGGCGAGC-3′, reverse, 5′-CAGGTCAGAACGAAGGTGCT-3′.

## Flow cytometry

VOrs were dissociated with Trypsin solution as described in single-cell dissociation. After resuspension in staining buffer, about $1 \times 10^6$ single cells for each group were incubated with Alexa 647-labeled CD31 antibody (1:1000 dilution, BD) for 30 min. The results were analyzed by using FlowJo software. Single cells isolated from BOrs were used as the negative control.

## LDL-uptake assay

hESC (D0) and VOrs (D4-D40) were washed with PBS three times and then incubated with 10 µg/ml Ac-LDL (Yeasen) in MV2 medium for 4 hr at 37°C. The samples were washed three times with PBS, before confocal imaging using ×20 objective lens.

## BBB-penetrating assay

fVBOrs were collected in 35 mm dishes and washed with PBS three times, followed by incubation with 5 µM angiopep (TAMRA-TFFYGGSRGKRNNFKTEEY) or control scrambled (TAMRA-GNYTSRFEREYGKFNKFGT) peptides in neuronal maturation medium for 3 hr at 37°C. Then, the organoids were washed with PBS, fixed in 4% PFA containing DAPI to stain the cell nuclei, and then imaged using ×1.25 objective lens.

## TEM analysis

Cultured fVBOrs were washed with DPBS (Life/Invitrogen) and cut into 1 mm × 1 mm small pieces. Firstly, samples were fixed with 4% PFA overnight and then were pre-fixed with 2.5% glutaraldehyde (SPI, USA), in PBS for 12 hr. After washing with PBS, samples were post-fixed with 1% OsO4 (TED PELLA) for 2 hr at 4°C. Then, dehydrated in an ascending gradual series (30–100% [v/v]) of ethanol and embedded in epoxy resin (Pon812 kit, SPI). The embedded samples were initially cut into about 500-nm-thick sections, inspected by stained with toluidine blue (Sinopharm), and finally sectioned into 70 nm by Leica EM UC7. Then sections were double-stained with uranyl acetate (SPI) and lead citrate (SPI), followed by observation with a TEM (Talos L120C) at an acceleration voltage of 80 kV.

## Whole-cell patch-clamp and organoid slice recording

Cultured BOrs and fVBOrs at D70 were collected and embedded in 3% agarose, and then sliced into 300-µm-thick sections in ice-cold cutting solution (100 mM glucose, 75 mM NaCl, 26 mM NaHCO$_3$, 2.5 mM KCl, 2 mM MgCl$_2$-6H$_2$0, 1.25 mM NaH$_2$PO$_4$-6H$_2$O, and 0.7 mM CaCl$_2$ in ddH$_2$O) by using a vibratome (Leica VT1200S). Then, slices were recovered in oxygenated (95% O$_2$ and 5% CO$_2$) artificial cerebrospinal fluid (ACSF, 124 mM NaCl, 25 mM NaHCO$_3$, 10 mM glucose, 4.4 mM KCl, 2 mM CaCl$_2$, 1 mM MgSO$_4$, and 1 mM NaH$_2$PO$_4$ in ddH$_2$O) for over 30 min. A piece of slice was transferred to the recording chamber within oxygenated ACSF, then voltage and current signal recording were performed using Axon 700B amplifier (Axon). The recording electrodes with 3–5 MΩ resistance were filled with the intracellular solution (112 mM Cs-Gluconate, 10 mM HEPES, 5 mM QX-314, 5 mM TEA-Cl, 3.7 mM NaCl, 2 mM MgATP, 0.3 mM Na$_3$GTP, and 0.2 mM EGTA in ddH$_2$O [adjusted to pH 7.2 with CsOH]) for sEPSC. For the voltage-induced current changes, the recording electrodes were filled with intracellular solution (120 mM K$^+$-glucose, 10 mM HEPES, 10 mM phosphocreatine, 5 mM NaCl, 2 mM MgATP, 0.2 mM EGTA, and 0.1 mM Na$_3$GTP in ddH$_2$O). In the voltage-clamp mode, the membrane potential was held at −60 mV for sEPSC recording. The currents responding to stimulating voltage pulses ranging from −80 mV to 60 mV (step 10 mV) were recorded.

## Single-cell dissociation and 10x Genomics chromium library construction

Organoids were dissociated using the methods as described previously (*Thomsen et al., 2016*). Briefly, 8–10 organoids were collected and washed with DPBS (Life/Invitrogen) and cut into small pieces, followed by the incubation with 2 ml trypsin solution (Ca$^{2+}$/Mg$^{2+}$-free HBSS [Life/Invitrogen] with 10 mM HEPES [Sigma-Aldrich], 2 mM MgCl$_2$, 10 µg/ml DNase Ⅰ [Roche], 0.25 mg/ml trypsin [Sigma-Aldrich], pH 7.6) for 30 min at 37°C. Then, the samples were quenched with 4 ml Quenching Buffer (440 ml Leibovitz L-15 medium [Thermo] with 50 ml ddH$_2$O, 5 ml 1 M HEPES [pH 7.3–7.4], 10 µg/ml DNase I, 100 nM TTX [Tocris], 20 µM DNQX [Tocris], and 50 µM DL-AP5 [Tocris], 5 ml 100× AntiAnti, 2 mg/ml BSA, 100 µg/ml trypsin inhibitor [Sigma-Aldrich]), subjected to centrifugation in 220

× *g* for 4 min at 4°C, and resuspended with 2 ml Staining Medium (440 ml Leibovitz L-15 medium with 50 ml ddH$_2$O, 5 ml 1 M HEPES [pH 7.3–7.4], 1 g BSA, 100 nM TTX, 20 µM DNQX, and 50 µM DL-AP5, 5 ml 100×AntiAnti, 20 ml 77.7 mM EDTA [pH 8.0]), filtered through a 40 micron cell filter (Falcon), centrifuged again in 220 × *g* for 4 min at 4°C, then the cells were resuspended in 5 ml DPBS with 1% BSA. Dissociated cells were resuspended at a concentration of 500 cells/µl. cDNA libraries were generated following the guidelines provided by 10x Genomics, Inc. Briefly, dissociated cells were partitioned into nanoliter-scale Gel Bead-In-Emulsions (GEMs), and then subjected to reverse transcription, cDNA amplification, and library construction, with individual cell and transcript barcoded. Additional reagent information is available in Appendix 1—key resources table.

## Single-cell RNA-seq data analysis

Cellranger software was used for mapping raw data to the human genome (version hg38 [v1.2.0]). Then, data were processed with Seurat (v3.0) under R (v3.5.2) environment. For quality control, cells expressing less than 200 genes or more than 7000 genes were removed, and genes expressed in less than three cells were excluded for the following analysis. Vvariance-stabilizing transformation (VST) was used for searching for the highly variable genes, and the top 2000 genes were chosen to do the downstream analysis. The cells were clustered and reduced into the UMAP space by PCA, with 1st to 15th principal components (PCs). DEGs in each cluster were identified by more than 1.25-fold change and p-value<.05 with Wilcoxon rank-sum test. GO enrichment analysis was carried out by 'ClusterProfiler' and org.Hs.eg.db in R software program with p-value<0.1 and false discovery rate (FSR) < 0.05 considered as statistical significance.

Monocle (v2.10.1) package was used to analyze the developmental trajectory of cell types. The monocle object was constructed from the Seurat object. After normalization and variance estimation, we calculated the mean and dispersion values and chose the genes whose mean expression were >0.1. Then, cells were dimensional reduced and clustered. DEGs with >1.25-fold change and p-value<0.05 (two-sided *t*-test) were used for downstream analysis. The cell trajectory plots were produced by running 'reduceDimension' and 'orderCell' function with defined option.

Human neocortical scRNA-seq data (*Polioudakis et al., 2019*) (phs001836) and mouse brain vascular scRNA-seq data (*He et al., 2018*; *Vanlandewijck et al., 2018*) (GSE98816) were downloaded for the correlation analysis. Pearson's correlation coefficient calculated by the intersect genes between different datasets was used for correlation analysis. DEGs were set with 1.25-fold change and p-value<0.05 by limma (v3.38.3) package. Additional software information is available in Appendix 1—key resources table.

## Acknowledgements

We are grateful to the Multi-Omics Core Facility, Molecular Imaging Core Facility, and Molecular and Cell Biology Core Facility at the School of Life Science and Technology, Cryo-Electron Microscopy Facility of ShanghaiTech University, and Core Facility of Center for Excellence in Brain Science and Intelligence Technology (CEBSIT) for providing technical support. We are grateful to Drs. Bo Peng and Yangfei Xiang for constructive suggestions during the course of this study, Dr. Min Zhang for the technical assistance on 10x Genomics chromium library construction, Dr. Xiaoming Li and Ms. Rui Wang for the assistance on 3D image reconstruction, and Ms. Linjie Li for the technical assistance on sample preparation for TEM. This study was partially supported by the National Key Research and Development Program of China (2021ZD0202500), National Natural Science Foundation of China (32130035 and 92168107 to ZGL, 31871034 to XCJ), the Frontier Key Project of the Chinese Academy of Sciences (QYZDJ-SSW-SMC025), Shanghai Municipal Science and Technology Projects (2018SHZDZX05, 201409001700), and National Key R&D Program of China (2017YFA0700500).

## Additional information

### Funding

| Funder | Grant reference number | Author |
|---|---|---|
| National Key Research and Development Program of China | 2021ZD0202500 | Zhen-Ge Luo |
| National Natural Science Foundation of China | 32130035 | Zhen-Ge Luo |
| National Natural Science Foundation of China | 92168107 | Zhen-Ge Luo |
| National Natural Science Foundation of China | 31871034 | Xiang-Chun Ju |
| Chinese Academy of Sciences Key Project | QYZDJ-SSW-SMC025 | Zhen-Ge Luo |
| Shanghai Municipal People's Government | 2018SHZDZX05 | Zhen-Ge Luo |
| Shanghai Municipal People's Government | 201409001700 | Zhen-Ge Luo |
| National Key Research and Development Program of China | 2017YFA0700500 | Xiang-Chun Ju |

The funders had no role in study design, data collection and interpretation, or the decision to submit the work for publication.

### Author contributions

Xin-Yao Sun, Data curation, Formal analysis, Investigation, Methodology, Validation, Visualization, Writing – original draft; Xiang-Chun Ju, Conceptualization, Data curation, Formal analysis, Funding acquisition, Investigation, Validation; Yang Li, Data curation, Investigation, Validation; Peng-Ming Zeng, Formal analysis, Validation; Jian Wu, Investigation, Validation; Ying-Ying Zhou, Jian Dong, Data curation, Investigation; Li-Bing Shen, Formal analysis; Yue-Jun Chen, Resources; Zhen-Ge Luo, Conceptualization, Funding acquisition, Project administration, Resources, Supervision, Writing – review and editing

### Author ORCIDs

Zhen-Ge Luo http://orcid.org/0000-0001-5037-0542

### Decision letter and Author response

Decision letter https://doi.org/10.7554/eLife.76707.sa1
Author response https://doi.org/10.7554/eLife.76707.sa2

## Additional files

### Supplementary files
• Transparent reporting form

### Data availability

Single cell RNA sequencing transcriptome data supporting this study have been deposited in NCBI Sequence Read Archive (SRA) repository (https://www.ncbi.nlm.nih.gov/sra) with accession number SRP338043 (VOR: SRR15992286; VOR2:SRR15992285).

The following dataset was generated:

| Author(s) | Year | Dataset title | Dataset URL | Database and Identifier |
| --- | --- | --- | --- | --- |
| Sun X-Y, X-C Ju, Zeng P-M, Wi J, Zhou Y-Y, Shen L-B, Dong J, Y-J Chen, Luo Z-G | 2021 | Generation of vascularized brain organoids to study neurovascular interactions | https://www.ncbi.nlm. nih.gov/sra/?term= SRP338043 | NCBI Sequence Read Archive, SRP338043 |

The following previously published datasets were used:

| Author(s) | Year | Dataset title | Dataset URL | Database and Identifier |
| --- | --- | --- | --- | --- |
| Vanlandewijck M, He L, Mäe M, Andrae J, Betsholtz C | 2017 | Single cell RNA-seq of mouse brain vascular transcriptomes | https://www.ncbi. nlm.nih.gov/geo/ query/acc.cgi?acc= GSE98816 | NCBI Gene Expression Omnibus, GSE98816 |
| Polioudakis D, de la Torre-Ubieta L, Langerman J, Elkins AG, Shi X, Stein JL, Vuong CK, Nichterwitz S, Gevorgian M, Opland CK, Lu D, Connell W, Ruzzo EK, Lowe JK, Hadzic T, Hinz FI, Sabri S, Lowry WE, Gerstein MB, Plath K, Geschwind DH | 2019 | A Single Cell Transcriptomic Analysis of Human Neocortical Development | https://www.ncbi.nlm. nih.gov/projects/gap/ cgi-bin/study.cgi? study_id=phs001836. v1.p1 | NCBI Gene Expression Omnibus, phs001836 |

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

# Appendix 1

### Appendix 1—key resources table

| Reagent type (species) or resource | Designation | Source or reference | Identifiers | Additional information |
|---|---|---|---|---|
| Antibody | GFP (chicken polyclonal) | Aves Lab | Cat# GFP-1020 | IF (1:1000) |
| Antibody | CD31 (mouse monoclonal) | Abcam | Cat# ab9498 | IF (1:300) |
| Antibody | PDGFRβ (goat polyclonal) | R&D | Cat# AF1042 | IF (1:200) |
| Antibody | αSMA (rabbit monoclonal) | Abcam | Cat# ab124964 | IF (1:500) |
| Antibody | DCX (goat polyclonal) | Santa Cruz | Cat# sc-8066 | IF (1:200) |
| Antibody | CLDN5 (mouse monoclonal) | Abcam | Cat# ab131259 | IF (1:500) |
| Antibody | IBA1 (rabbit monoclonal) | Wako | Cat# 019-19741 | IF (1:500) |
| Antibody | PAX6 (sheep polyclonal) | R&D | Cat# AF8150 | IF (1:500) |
| Antibody | p-VIM (mouse monoclonal) | MBL | Cat# D076-3 | IF (1:1000) |
| Antibody | DLL4 (rabbit polyclonal) | Abcam | Cat# ab7280 | IF (1:500) |
| Antibody | EPHB4 (rabbit monoclonal) | Cell Signaling | Cat# 14960 | IF (1:500) |
| Antibody | KI67 (mouse monoclonal) | BD | Cat# 550609 | IF (1:1000) |
| Antibody | TBR2 (mouse monoclonal) | R&D | Cat# AF6166 | IF (1:400) |
| Antibody | TUJ1 (chicken polyclonal) | Abcam | Cat# ab41489 | IF (1:1000) |
| Antibody | TBR1 (rabbit polyclonal) | Abcam | Cat# ab31940 | IF (1:500) |
| Antibody | CTIP2 (rat monoclonal) | Abcam | Cat# ab18465 | IF (1:500) |
| Antibody | SATB2 (rabbit polyclonal) | Abcam | Cat# ab69995 | IF (1:400) |
| Antibody | PSD95 (rabbit monoclonal) | Cell Signaling | Cat# 3450S | IF (1:500) |
| Antibody | ZO-1 (rabbit monoclonal) | Abcam | Cat# ab221547 | IF (1:500) |
| Antibody | GLUT1 (rabbit monoclonal) | Abcam | Cat# ab115730 | IF (1:500) |
| Antibody | p-Glycoprotein (rabbit monoclonal) | Abcam | Cat# ab170904 | IF (1:500) |
| Antibody | Cleaved-CASPASE3 (rabbit polyclonal) | Cell Signaling | Cat# 9661L | IF (1:500) |
| Antibody | TMEM119 (rabbit polyclonal) | Abcam | Cat# ab185333 | IF (1:500) |
| Antibody | TREM2 (rabbit monoclonal) | Abcam | Cat# ab209814 | IF (1:500) |

*Appendix 1 Continued on next page*

*Appendix 1 Continued*

| Reagent type (species) or resource | Designation | Source or reference | Identifiers | Additional information |
|---|---|---|---|---|
| Antibody | Human-Nuclei (mouse monoclonal) | Millipore | Cat# MAB1281 | IF (1:500) |
| Antibody | Anti-Hu CD31 Alexa 647 WM59 50Tst | BD | Cat# 561654 | Flow cytometry (1:1000) |
| Peptide, recombinant protein | Hu Recom bFGF | STEMCELL | Cat# 78003 | |
| Peptide, recombinant protein | Hu Recom VEGF | STEMCELL | Cat# 78073 | |
| Peptide, recombinant protein | Hu Recom BMP4 | R&D | Cat# 314BP | |
| Peptide, recombinant protein | Insulin | Sigma-Aldrich | Cat# I9278 | |
| Peptide, recombinant protein | Human Dil-acetylated low-density lipoprotein | Yeasen | Cat# 20606ES76 | |
| Chemical compound, drug | Y27632 | STEMCELL | Cat# 72304 | |
| Chemical compound, drug | Dorsomorphine | Tocris | Cat# 3093/10 | |
| Chemical compound, drug | A83-01 | Tocris | Cat# 2939/10 | |
| Chemical compound, drug | CHIR99021 | Selleck | Cat# S1263 | |
| Chemical compound, drug | LDN-193189 2HCL | Selleck | Cat# S7507 | |
| Chemical compound, drug | SB431542 | Selleck | Cat# S1067 | |
| Chemical compound, drug | TTX | Tocris | Cat# 1069 | |
| Chemical compound, drug | DNQX | Tocris | Cat# 0189 | |
| Chemical compound, drug | DL-AP5 | Tocris | Cat# 3693 | |
| Chemical compound, drug | Trypsin inhibitor | Sigma-Aldrich | Cat# T6522 | |
| Chemical compound, drug | PLX5622 | MCE | Cat# HY-11415 | |
| Chemical compound, drug | LPS | MCE | Cat# HY-D1056 | |
| Commercial assay, kit | SYBR Green PCR mix | Bimake | Cat# B21702 | |
| Commercial assay, kit | RNeasy Plus Micro Kit | QIAGEN | Cat# 74034 | |
| Commercial assay, kit | GoScript Reverse Transcription Kit | Promega | Cat# A5001 | |
| Commercial assay, kit | Pon812 812 kit | SPI | Cat# GS02660 | |

*Appendix 1 Continued on next page*

*Appendix 1 Continued*

| Reagent type (species) or resource | Designation | Source or reference | Identifiers | Additional information |
|---|---|---|---|---|
| Commercial assay, kit | Toluidine blue | Sinopharm | Cat# XW65860453 | |
| Commercial assay, kit | Uranyl acetate | SPI | Cat# GS02624 | |
| Commercial assay, kit | Lead citrate | SPI | Cat# GP19314 | |
| Commercial assay, kit | Single Cell Reagent Kits | 10x Genomics | N/A | |
| Software, algorithm | Cellranger | 10x Genomics | https://support.10xgenomics.com/single-cell-gene-expression/software/overview/welcome | |
| Software, algorithm | Seurat (v3) | *Macosko et al., 2015* | https://satijalab.org/seurat/ | |
| Software, algorithm | R (v3.5.2) | N/A | https://www.r-project.org/ | |
| Software, algorithm | clusterProfiler (v3.10.1) | | http://bioconductor.org/packages/release/bioc/html/clusterProfiler.html | |
| Software, algorithm | Limma (v3.38.3) | | http://bioconductor.org/packages/release/bioc/html/limma.html | |
| Software, algorithm | Monocle (v2.10.1) | | http://cole-trapnell-lab.github.io/monocle-release/ | An analysis toolkit for single-cell RNA-seq. |
| Software, algorithms | Fiji | N/A | https://fiji.sc | |
| Software, algorithm | Angiotool (v 0.6a) | *Zudaire et al., 2011* | http://angiotool.nci.nih.gov | |
| Software, algorithm | Reference Transcriptome for GRCh38 (v1.2.0) | N/A | https://genome.ucsc.edu/ | |
| Other | Heparin | Sigma-Aldrich | Cat# H3393 | Section 'Generation of human brain organoid' |
| Other | Lipidure | NOF CORPORATION | Cat# CM5206 | Section 'Generation of human brain organoid' |
| Other | Antibiotic-Antimycotic | Gibco | Cat# 15240096 | Section 'Generation of human brain organoid' |
| Other | Matrigel hESC-Qualified Matrix | BD-Biocoat | Cat# 354277 | Section 'hESCs culture' |
| Other | Matrigel growth factor reduced (GFR) basement membrane matrix | BD-Biocoat | Cat# 354230 | Section 'Generation of human brain organoid' |
| Other | STEMdiff APEL2 Medium | STEMCELL | Cat# 05270 | Section 'Generation of human vessel organoid' |
| Other | Endothelial cell growth medium MV2 | PromoCell | Cat# C-22022 | Section 'Generation of human vessel organoid' |
| Other | mTeSR1 | STEMCELL | Cat# 85850 | Section 'hESCs culture' |
| Other | ReLeSR | STEMCELL | Cat# 05872 | Section 'hESCs culture' |
| Other | Accutase | STEMCELL | Cat# 07920 | Section 'Generation of human brain organoid' |
| Other | O.C.T | Sakura | Cat# 4583 | Section 'Immunofluorescence' |

*Appendix 1 Continued on next page*

*Appendix 1 Continued*

| Reagent type (species) or resource | Designation | Source or reference | Identifiers | Additional information |
|---|---|---|---|---|
| Other | BSA | Sigma-Aldrich | Cat# V900933 | Section 'Immunofluorescence' |
| Other | TritonX-1000 | Sigma-Aldrich | Cat# T8787 | Section 'Immunofluorescence' |
| Other | Neurobasal | Life/Invitrogen | Cat# 21103049 | Section 'Generation of human brain organoid' |
| Other | N2 supplement | Life/Invitrogen | Cat# 17502048 | Section 'Generation of human brain organoid' |
| Other | B27 supplement without vitamin A | Life/Invitrogen | Cat# 12587010 | Section 'Generation of human brain organoid' |
| Other | B27 supplement | Life/Invitrogen | Cat# 17504044 | Section 'Generation of human brain organoid' |
| Other | DPBS | Life/Invitrogen | Cat# 14190144 | Section 'hESCs culture' |
| Other | HBSS | Life/Invitrogen | Cat# 14175069 | Section 'Single-cell dissociation and 10x Genomics chromium library construction' |
| Other | HEPES | Sigma-Aldrich | Cat# H4034 | Section 'Single-cell dissociation and 10x genomics chromium library construction' |
| Other | DNase I | Roche | Cat# 10104159001 | Section 'Single-cell dissociation and 10x Genomics chromium library construction' |
| Other | Knockout Serum Replacer | Gibco | Cat# 10828028 | Section 'Generation of human brain organoid' |
| Other | MEM-NEAA | Gibco | Cat# 11140050 | Section 'Generation of human brain organoid' |
| Other | GlutaMAX | Gibco | Cat# 35050061 | Section 'Generation of human brain organoid' |
| Other | β-Mercaptoethanol | Sigma-Aldrich | Cat# M3148 | Section 'Generation of human brain organoid' |
| Other | Bovine pancreatic trypsin | Sigma-Aldrich | Cat# 6502 | Section 'Single-cell dissociation and 10x Genomics chromium library construction' |
| Other | Leibovitz L-15 medium | Thermo | Cat# 11415064 | Section 'Single-cell dissociation and 10x Genomics chromium library construction' |
| Other | DMEM/F12 | Life/Invitrogen | Cat# 10565018 | Section 'hESCs culture' |

