## [Editor Report]

This article puts forward a new approach to generate vascularized brain organoids. The novelty of their approach lies in the simultaneous production of vessel-like networks and brain-resident microglia immune cells in a single organoid, and data demonstrating that the vessels are patent to allow fluid flow when pressurized fluid is delivered to the vascular tube. The fusion of brain and vessel organoids resulted in robust engraftment of vessel-like structures and microglia around ventricular zone-like structures, correlating with increased neuronal progenitors.

---

## [Decision Letter]

**Decision letter after peer review:**

Thank you for submitting your article "Generation of Vascularized Brain Organoids to Study Neurovascular Interactions" for consideration by *eLife*. Your article has been reviewed by 2 peer reviewers, and the evaluation has been overseen by a Reviewing Editor and Jeannie Chin as the Senior Editor. The reviewers have opted to remain anonymous.

Essential revisions

1) Why is there a decrease in GFP+, CD31+ cell percentage in later stages of VOrs (D40) compared to earlier stages in flow cytometry analysis in Figure 1 – Figure Supp. 1C? This may indicate that most of cells in VOrs were still early mesodermal cells or undifferentiated. Otherwise, endothelial induction efficiency should be enhanced.

2) SATB2 staining in Figure 4 – Figure Supp. 1E seems to label neural progenitors in VZ-like region. SATB2 is expressed in mature neurons but not in the progenitors. Please confirm the reliability of this staining.

3) Related to figure 5F, the authors described that large amount of IBA1+GFP+ MG-like cells were found in fVBOrs. However, large amount of IBA1+GFP- cells were also observed in fvBOrs. Thus MG-like cells might be activated upon vascularization. The authors should address this possibility.

---

## [Author Response]

Essential revisions1) Why is there a decrease in GFP+, CD31+ cell percentage in later stages of VOrs (D40) compared to earlier stages in flow cytometry analysis in Figure 1 – Figure Supp. 1C? This may indicate that most of cells in VOrs were still early mesodermal cells or undifferentiated. Otherwise, endothelial induction efficiency should be enhanced.

The seemingly decrease in CD31^+^GFP^+^ cell percentage is most likely due to the appearance of other cell types in later culture stages (D40), such as fibroblasts, smooth muscle cells, pericytes, et al. In line with this prediction, single cell RNA-seq (scRNA-seq) analysis showed that in D40 VOrs, fibroblasts occupied around half of all cell types and endothelial cells only accounted for 10% (Figure 2A and Figure 2-supplement 1B). The percentage of endothelial cells indicated by scRNA-seq and flow cytometry was similar. The low percentage of MKI67^+^ proliferative vascular progenitor and appearance of various cell types indicate efficient differentiation induction.

2) SATB2 staining in Figure 4 – Figure Supp. 1E seems to label neural progenitors in VZ-like region. SATB2 is expressed in mature neurons but not in the progenitors. Please confirm the reliability of this staining.

Sorry for the confusion. The SATB2 signal in VZ-like region was seen in rare cases, and this might be due to the presence of cell types in intermediate states. To avoid the confusion, we have replaced it with a more representative image (see revised Figure 4-supplyment 1E).

3) Related to figure 5F, the authors described that large amount of IBA1+GFP+ MG-like cells were found in fVBOrs. However, large amount of IBA1+GFP- cells were also observed in fvBOrs. Thus MG-like cells might be activated upon vascularization. The authors should address this possibility.

Thanks to the Reviewer for this interesting point. Based on the brain RNA-seq database of various cell types in mouse and human (Zhang Y, Chen K, et al., J Neurosci 2014, 34, 11929-47), VEGFR1 (encoded by FLT1 gene) is highly expressed in human microglia. It is very possible that the residue microglia in BOrs can sense the signals, such as VEGF, from the invading vessels to be motivated and activated. This point has been mentioned in the revised manuscript (lines 310-314).